# Isolation of Two Bacterial Species from Argan Soil in Morocco Associated with Polyhydroxybutyrate (PHB) Accumulation: Current Potential and Future Prospects for the Bio-Based Polymer Production

**DOI:** 10.3390/polym13111870

**Published:** 2021-06-04

**Authors:** Amina Aragosa, Valeria Specchia, Mariaenrica Frigione

**Affiliations:** 1Department of Biological and Environmental Sciences and Technologies, University of Salento, 73100 Lecce, Italy; a.aragosa@aui.ma (A.A.); valeria.specchia@unisalento.it (V.S.); 2School of Science and Engineering, Al Akhawayn University, Ifrane 53000, Morocco; 3Department of Innovation Engineering, University of Salento, 73100 Lecce, Italy

**Keywords:** bio-based polymers, polyhydroxybutyrate, PHB-producing bacteria, argan soil, *Argania spinosa*

## Abstract

The environmental issues caused by the impacts of synthetic plastics use and derived wastes are arising the attention to bio-based plastics, natural polymers produced from renewable resources, including agricultural, industrial, and domestic wastes. Bio-based plastics represent a potential alternative to petroleum-based materials, due to the insufficient availability of fossil resources in the future and the affordable low cost of renewable ones that might be consumed for the biopolymer synthesis. Among the polyhydroxyalkanoates (PHA), the polyhydroxybutyrate (PHB) biopolymer has been synthesized and characterized with great interest due to its wide range of industrial applications. Currently, a wide number of bacterial species from soil, activated sludge, wastewater, industrial wastes, and compost have been identified as PHB producers. This work has the purpose of isolating and characterizing PHB-producing bacteria from the agricultural soil samples of *Argania spinosa* in the south region of Morocco where the plant species is endemic and preserved. During this research, four heat-resistant bacterial strains have been isolated. Among them, two species have been identified as endospore forming bacteria following the Schaffer-Fulton staining method with Malachite green and the Methylene blue method. Black intracellular granules have been appreciated in microscopy at 100× for both strains after staining with Sudan black B. The morphological and biochemical analyses of the isolates, including sugar fermentation and antibiotic susceptibility tests, preliminarily identified the strains 1B and 2D1 belonging to the genus *Serratia* and *Proteus*, respectively.

## 1. Introduction

The problems caused by the persistent production and use of synthetic plastics, the accumulation of their wastes, and impossibility to be decomposed have become one of the major worldwide environmental issues [1]. In the last decades, more attention has been addressed to pursue sustainability and to opt to eco-friendly materials which preserve the environment and its biodiversity. On this perspective, the use of petrochemical plastics, produced starting from a non-renewable resource, has been discouraged since they are not biodegradable, toxic when accumulated into landfills and incinerated, and they are potentially harmful for the environment when entering aquatic and terrestrial ecosystems. Biobased plastics are gaining increasing attention since they can be produced from renewable resources, reducing the depletion of petroleum and the amount of gas emissions [2].

To overcome the problems of conventional plastics, polyhydroxyalkanoates (PHAs) represent a promising competitive material due to their biodegradable and biocompatible properties [3], but also because, similar to conventional plastics PHAs are elastic, resistant to chemicals, have low hardness, and poor mechanical properties [4]. The biopolymer PHAs are produced by bacteria, particularly soil microorganisms following sugar and lipids fermentation. These biopolyesters, consisting of a variety of structures depending on the nature of the producers, growth media, culture and extraction parameters, are primarily used for packaging purposes. Furthermore, PHAs can be developed into biocomposites and copolymers by combining the PHAs to various additives to improve its physio-chemical properties and expanding its uses to the medical field, tissue engineering, and electronic industry. PHAs are suitable alternatives to synthetic plastics since they are highly impermeable to oxygen permeation preventing the oxidative spoilage of the packed products. Moreover, their biosynthetic and biodegradability properties represent further advantages suitable for packaging improvement and the preservation of food quality [5].

Nowadays, the polyhydroxybutyrate (PHB) is one of the most studied and characterized biopolymer in the PHA family, synthesized by either Gram-positive or Gram-negative microorganisms [6]. PHB is a linear homopolymer of D-3-hydroxybutyric acid with 100% R configuration [7], similar to polypropylene (PP), due to its good resistance to moisture and the possibility to be locally crystallized in helical structures, such as isotactic PP. PHB was identified for the first time by Lemoigne in 1925 [2]. It is different from other bio-based polymers since it is hydrophobic, resistant to UV lights and hydrolytic degradation, permeable to oxygen, not resistant to acids and bases, and soluble in chloroform and other chlorinated chemical agents. Due to all these good mechanical and thermal properties, PHB captured the attention of industrial productions [8]. PHBs have been used mainly for packaging, in agriculture, food industry, and also in medical and pharmaceutical fields [9]. Moreover, PHB can be degraded, aerobically and anaerobically, by a biodegradation process [2].

The PHB molecular weight, that can vary from 50,000 to 1,000,000, depends on the microbial producer, the culture optimization conditions for the bacterial growth and the biosynthesis, and its extraction method [7]. Thammasittirong et al. [7] reported that not only the species and the substrate used can influence the quality of PHB synthesized, but also the culture optimization parameters, such as nutrients and their proportions, pH, and temperature. They reported how the *Bacillus thuringiensis*, strain B417-5, exhibits high efficiency production on a large scale using sugarcane, at a neutral pH, incubating for 48 h at 37 °C and shaking at 100 rpm [9]. The 70 to 80% of the PHB production costs account for the carbon sources, which is why it is important to adopt wastes rich in carbon to afford the large-scale production of this polymer. Recent studies showed that also cow dung rich in carbon, proteins, and nitrogen represent an effective cheap resource for the PHB production. In their recent study, Chandani Devi et al. [10] used the *Bacillus pumilus* strain H9 and for the first time the cheap, easily available, and nutritionally rich cow dung waste as the sole source of carbon. The maximum amount of PHB was produced by consuming 5 g/L of cow dung, after 48 h of incubation at pH 7 [10]. However, due to the high cost of production of the biopolymer on a large scale, this research considers the following aspects: Improvement of the fermentation and extraction process; development of new recombinant bacterial strains; and use of inexpensive carbon sources as substrates [11].

Among the PHA family, the PHB is a biodegradable polymer synthesized from carbon assimilation by some bacterial species that accumulate the biopolymer into intracellular lipid granules during harsh environmental conditions as a form of storage molecule [8]. A particular strain of *Pseudomonas extremaustralis* isolated in Antarctica owns the PHB gene effective to stress conditions including the extremely low temperature of the region [12]. A wide variety of microorganisms have been isolated from soil [2], wastewater, industrial effluent, domestic sewage, and dairy wastes [13]. Numerous studies have been conducted to isolate efficient PHB-producing bacteria from soil samples. In their study, Lathwal et al. [14] isolated a total of 194 PHB producers from rizhospheric crop soils of wheat, sugarcane, and mustard in India. Among all the strains belonging to the genera *Bacillus*, *Lysinibacillus*, *Clostridium*, and *Klebsiella*, few promising isolates were identified as novel strains exhibiting the potential to produce the biopolymer in a large scale [14]. Moreover, other PHB isolates were identified in paddy soil, sunflower soil, and particularly those isolated from red soil produced a higher final yield of PHB [8]. Again, PHB producers were isolated from soil samples in different regions of Saudi Arabia, identified as *Bacillus* sp. and characterized as efficient producers in culture optimization conditions, such as media composition, aeration, incubation time and temperature, pH, carbon and nitrogen source [15]. Most of the identified species are responsible for the biosynthesis and biodegradation of the biopolymer due to the enzymatic activity of synthetases and depolymerases [8]. The biosynthesis of bacterial PHB is a set of catalyzed chemical reactions that convert the initial carbon source, generally a sugar, into the PHB biopolymer by the action of three main enzymes: The â-ketothiolase that converts the Acetyl-CoA into Acetoacetyl-CoA, acetoacetyl-CoA reductase which transforms the Acetoacetyl-CoA into the (R)-3-hydroxybutyryl-CoA (PhaB), and the PHB synthase which finally triggers the reaction to the synthesis of PHB [16]. During imbalanced growth conditions, the level of acetyl-CoA increases while the level of Co-A decreases. This triggers the enzyme â-ketothiolase to start the three-step process for the synthesis of PHB. On the other hand, when the level of Co-A increases the first enzyme, â-ketothiolase, is inhibited and the synthesis of the biopolymer stops. This activation and deactivation process of the PHB synthesis is regulated by the enzyme â-ketothiolase which is a key catalyst of the synthesis [17]. Examples of PHB producers belong to the species *Azotobacter*, *Bacillus*, and *Pseudomonas*. Again, many bacterial species accumulate the biopolymer by consuming agricultural carbon sources, such as corn wastes, rice straw, soybean, mustard, sesame, corn oil, frying oil, and all types of wastes that rather than being degraded can be used to produce the biopolymer by reducing the cost of disposal and for the biosynthesis on a large scale at low-cost production [16].

PHB producing bacteria have been classified into two main groups according to the nutrient medium composition for bacterial growth. The *Ralstonia eutropha* and *Pseudomonas oleovorans* belong to the first group of bacteria that requires a nitrogen limitation for growth, while *Bacillus turingiensis*, *Bacillus subtilis*, and *Alcaligenes latus* depend on the carbon source for the accumulation of PHB but do not depend on the nitrogen or phosphorous limitations [13]. For instance, *Ralstonia eutropha* (*Alcaligenes eithrofus*) in optimal culture conditions, can accumulate up to 80% of dry weight, while the most studied *Cupriavidus necator* showed the capacity to synthetize high amounts of the biopolymer (80% *w*/*w*) from easy carbon sources such as acetic acid, oleic acid, glycerol, sucrose, and lactic acid [16]. PHB production using *Cupriavidus necator* has been characterized in several studies. One promising work suggested the use of volatile fatty acids obtained from the anaerobic digestion of municipal waste paper. This work, conducted by Battashi et al. [3], not only suggests that PHA might be produced at a final yield of 53.50%, but it also reduces the high-cost production since no pretreatments, enzymatic digestion, detoxification of the final product, and additional nutrients are required [3]. On the same promising results, other species such as *Azotobacter vinelandii* can accumulate up to 75% of PHB during its exponential growth, and *Protomonas extorquens* can synthetize the PHB in a fermentation batch with methanol. *Rhodococcus equi* has its maximum amount of PHB by consuming crude palm kernel oil as the sole source of carbon [18], while *Bacillus*, *Pseudomonas*, *Micrococcus,* and *Veillonella* strains demonstrated the highest production at 24 h of fermentation using glucose as the sole source of carbon [19]. *Bacillus* sp. produce a high amount of PHB which can hardly be extracted due to the spontaneous sporulation of the producer. However, bacillus strains are able to synthesize short and medium PHB chains, with different molecular weights, used to produce PHB copolymers (PHV, PHH) that could be used to solve some of the mechanical and thermal limits of the pure PHB [20]. More innovative molecular techniques are looking to recombinant *Escherichia coli*, to induce more bacterial strains to produce the biopolymer [2].

The present work was designed to identify the capability of isolated bacteria from rizhospheric argan soil to accumulate the biopolymer polyhydroxybutyrate. In this study, we investigate the potential presence of microorganisms able to synthesize the biopolymer PHB from *Argania spinosa* crop soil in an unique environmental area of Morocco where this species is endemic and preserved, and we preliminarily characterize the morphological, biochemical, and antibiotic susceptibility properties of the isolates. Our work was conducted as follows: (i) collecting, heating, and diluting rizhospheric soil samples to select thermoresistant bacteria; (ii) investigating among the thermoresistant bacteria the presence of endospore-forming bacteria by the Schaeffer-Foulton staining method specific for endospores; (iii) investigating among the endospore-forming bacteria the presence of PHB granules by the Sudan black staining method specific for PHB producers; (iv) characterizing morphological and Gram staining properties of the PHB positives at the compound light microscope; (v) examining biochemical properties of the PHB producers by the API20E system and supplemental sugar fermentation tests; (vi) examining the antibiotic susceptibility characteristics of the isolated PHB-producing bacteria. While the work conducted in (i), (ii), and (iii) aims to isolate PHB-producing bacteria, the further investigations in (iv), (v), and (vi) aim to preliminarily identify the genus to which the isolates belong to. 

## 2. Materials and Methods

### 2.1. Soil Collection and Isolation of Heat Resistant Bacteria 

From the agriculture crops of argan trees, in the southwest region of Morocco where the species *Argania spinosa* is endemic, rhizospheric soil samples were extracted using soil probes disinfected with ethanol 70% and collected into sterile plastic tubes (Figure 1a,b). The distinctive physio-chemical parameters of the locations might play a role to best suit the growth of PHB synthesizers. Indeed, some of the parameters considered were the presence or absence of water, domestic pollutants, grazing, competitive growth, and geomorphology of the soil. All soil samples were collected and preserved in vials using sterile conditions, stored at 4 °C for 3 weeks for the bacteria isolation. Afterwards, all soil samples were stored at 4 °C for the isolation of heat resistant bacteria. Before proceeding with the bacterial isolation, each soil sample was pretreated as reported below. One gram of each soil sample was dispersed in 10 mL of sterile water, homogenized at 200 rpm for 3 h and heated at 80 °C for 10 min to select only heat resistant bacteria. Subsequently, the samples were serially diluted with sterile water until 10^−8^ and plated on nutrient agar composed of peptone 5 g/L, yeast extract 3 g/L, sodium chloride 5 g/L, glucose 1 g/L, agar 18 g/L, in 1 L of distilled water, at pH 7. Plates were incubated at 30 °C for 48 h and all grown bacteria were isolated in modified agar containing beef extract (0.3%), peptone (0.5%), sodium chloride (0.8%), glucose (1%), and agar (1.5%) [19]. All isolated bacteria were examined for the presence of endospores with the Schaeffer-Fulton staining methods. 

### 2.2. Screening for Endospore Forming Bacteria

The Schaeffer-Fulton methods for staining endospores with Malachite Green [21] and Methylene Blue [22] have been implemented to identify endospore forming bacteria. After fixation of the microorganisms on the glass slide, both staining procedures have been performed by covering the specimens with blotting paper saturated with the respective staining solution on a steaming bath. After washing the specimens with distilled water, specimens were counter stained with Safranin (2.5% in distilled water) and observed at the compound light microscope with immersion oil at 100×. In endospore forming bacteria, the warm steam allows the Malachite Green (0.5%) and the Methylene Blue (0.5% at pH 7) dyes to penetrate the alkaline cytosol and the endospore membrane. The vegetative cells appear red, while the endospores show a bright green or blue color depending on whether the Malachite Green or the Methylene Blue were used [22]. All bacteria identified as endospore forming bacteria were, afterwards, examined for the production of PHB granules with the Sudan black staining. 

### 2.3. Screening for PHB-Producing Bacteria

#### 2.3.1. Sudan Black Staining on Petri Dishes

The bacteria isolated as endospore forming positive were examined for the accumulation of PHB by staining the bacterial smear directly on the petri dish using an alcoholic solution of Sudan black (0.3%). After lasting the stain for 20 min, the petri dishes were washed with 96% ethanol to remove the excess of dye and observed with the naked eye [1]. Smears that appeared dark blue-black were selected as PHB producers and examined, once again, with Sudan black staining on the glass slide for microscopic investigation.

#### 2.3.2. Sudan Black Staining on Glass Slides

All smears that resulted as positive were detected for the production of PHB granules using the Sudan black B solution (30%). The dye was prepared by mixing 0.3 mg of the solid dye with 100 mL of 70% ethanol. Smears of the positive bacteria were prepared and heat-fixed on glass slides. Specimens were stained with the Sudan black B solution for 15 min, washed with xylene, and counter stained with Safranin (5% in distilled water) for 1 min before being observed under the microscope with immersion oil, at 100× [23]. Cells containing blue-black cytoplasmic granules were considered PHB producers, preserved in 2% glycerol vials for preservation and further analysis. 

### 2.4. Morphology and Biochemical Characterization of the Isolates

#### 2.4.1. Morphology and Gram Staining

Bacteria identified and preserved as PHB producers were studied morphologically to characterize the colony and the cellular properties. The shape, margin, color, elevation, size, surface, texture, and arrangement are the characteristics analyzed to determine the colony morphology on nutrient agar plates [24], while the shape, color, arrangement, size [25,26], and Gram staining [27] were observed to determine the cellular characteristics of the isolates at the microscope with immersion oil at 100×. 

#### 2.4.2. Biochemical Characterization

After completing the morphological analysis, the selected bacteria were identified by performing the following biochemical tests: Beta galactosidase, arginine dehydrolase, lysine decarboxylase, ornithine decarboxylase, citrate utilization, H_2_S production, urea hydrolysis, tryptophane deaminase, indole production, gelatin liquefaction, and oxidase performed using the API 20E system for Enterobacteriaceae according to the manufacturer instructions (Biomerieux). Additionally, various carbohydrate fermentation tests (glucose, mannose, rhamnose, melibiose, sucrose, arabinose, lactose, fructose, maltose, galactose, starch, dulcitol) were conducted to study the bacterial sugar utilization. The isolates were inoculated separately in a medium containing each of the sugars previously listed and incubated at 37 °C for 48 h. Estimations of the bromothymol blue indicator color changes were used to determine the positive and negative fermenters [26].

#### 2.4.3. Antibiotic Susceptibility

The antibiotic susceptibility of the Kirby-Bauer test protocol [28] was performed to study the resistance and susceptibility of the PHB producers to the following antibiotics for further characterization: Amoxicillin (2 μg), cefepime (30 μg), ampicillin/sulbactam (20 μg), flumequine (30 μg), penicillin G (10 μg), imipenem (10 μg), roxithromycin (15 μg), nitroxoline (20 μg), floxapen (5 μg), colistin sulphate (50 μg), levofloxacin (5 μg), gentamicin (10 μg), chloramphenicol (30 μg), teicoplanin (30 μg), bacitracin (10 μg), amoxicillin/clavulanic acid (30 μg), and kanamycin (30 μg). First, bacterial isolates were grown in a Luria-Bertani nutrient broth for 24 h at 37 °C. The inoculum was then transferred using a swab to the Muller Hinton agar plates by streaking on the entire agar surface for an even distribution of the inoculum. The appropriate antimicrobial-impregnated disks were placed on the surface of the agar one at a time using forceps. The inverted plates were incubated for 18 h at 35 °C before measuring the zones of inhibition for susceptibility interpretation using the published CLSI guidelines [28].

## 3. Results and Discussion

### 3.1. Soil Collection and Isolation of Heat Resistant Bacteria 

To investigate the presence of bacteria able to synthetize the polyhydroxybutyrate, rizospheric soil samples were collected from argan tree (*Aragania spinosa)* crops. From the soil samples collected and examined, four bacterial species were identified as thermoresistant. The isolates named strain 1B, 3B, 4B were isolated from an argan crop located in proximity of urban areas contaminated by wastewater and garbage, while the strain 2D1 was isolated from a crop used for extensive grazing exploitation. Recent studies already demonstrated that polluted samples, including sewage soils [25], cardboard industrial wastes [29], cattle rumen fluid [11], cow dung, slurry, and tobacco dust [19] represent a carbon rich container for the isolation of high productive PHB synthesizers. 

### 3.2. Screening for Endospore Forming Bacteria

The Schaeffer-Fulton staining method using Malachite green for a conventional staining, and Methylene blue for an alternative approach, were used to observe the presence of lipophilic endospores. Among the four thermoresistant isolates, only two strains, 1B and 2D1, showed a green coloration for the retainment of the Malachite green which, due to its lipophilic activity, passively cross the cell and the granule membranes (Figure 2a,b). Moreover, among the four thermoresistant isolates, also with the alternative Methylene blue staining, only 1B and 2D1 strains revealed the presence of dark blue endospores. Several works reported how the alternative Methylene blue stain can substitute the Malachite green for the endospore staining, due to its chemical properties to stain negatively charged molecular components, including nucleic acids and liposoluble membranes. Indeed, a study was reported to show the effectiveness of different concentrations and pH levels of Methylene blue solutions to observe endospores in the species *Bacillus subtilis* and *Clostridium tetani.* The study proved that the optimal staining concentration and pH level for *Bacillus subtilis* and *Clostridium tetani* is 0.5% at pH 12 and 0.5% at pH 11, respectively [22]. In this work, the optimal concentration and pH level that best revealed the presence of endospores with Methylene blue was 0.5% at pH 12 for both strains, 1B and 2D1, as reported in Figure 2c,d. 

### 3.3. Screening for PHB-Producing Bacteria

#### Sudan Black Staining on Petri Dish and Glass Slide

In our study, all four isolates were tested for microscopic and plate screening for PHB production using Sudan black B staining. Of the four isolates, only the two strains, named 1B and 2D1, were identified as positive at the Sudan black B, therefore able to accumulate PHB in the lipidic granules. In Figure 3a,b, the staining of the bacterial strains 1B and 2D1 is reported with the Sudan black B staining directly on petri dish using a smear growth culture of the respective isolates. The dark blue and black color of the smears were taken as PHB positive. Additionally, the same staining using the Sudan black B solution was implemented for microscopic screening, by following the staining method on a glass slide. Figure 4a,b indicates the presence of black intracellular granules for both strains, which confirm the positivity of the previous plate staining and the necessity to proceed with further investigations to characterize the isolated PHB positives. It is reported in the literature that over 90 bacterial genera have been identified as PHB producers. Among all the isolated and identified species, many colonize soil environments, and few have been isolated from waste water, sludge ecosystems, industrial wastes, and animal manure [9]. 

### 3.4. Morphological and Biochemical Characterization of the Isolates

#### 3.4.1. Morphology and Gram Staining

The morphological characteristics of 1B and 2D1 strains, which resulted as positive with the Sudan black staining for the presence of PHB granules, were observed in modified agar pure cultures after incubation at 37 °C for 24 h. Their morphological characteristics can be appreciated in Table 1 and Figure 5.

At the naked eye, both PHB positive strains formed regular white colonies, translucent, with a smooth surface. However, while strain 1B (Figure 5a) formed viscid clustered colonies with a diameter between 3 to 5 mm, the strain 2D1 (Figure 5b) grew into isolated dry colonies with a smaller diameter between 0.5 to 1 mm. Moreover, both PHB producers were identified as Gram-negative bacteria, and the bacterial red color appreciated at 100× magnification power can be observed in Figure 6. 

Among the biopolymers, the PHB is the most characterized and synthesized by different Gram-positive Bacillus sp., such as *Bacillus megaterium, B. subtilis,* and *B. thuringiensis*, but also by Gram-negative bacteria such as *Cupriavidus necator*, *Azotobacter vinelandii*, and *Pseudomonas mendocina* [9]. Unlike Gram-positive species, the Gram-negative species produce endotoxins in the outer membrane of the lipophilic granules. Unfortunately, this cellular mechanism represents a disadvantage not conducive for the medical field applications of PHB synthesized by Gram-negative bacteria, since the high cost for the product purification makes a large-scale production difficult [20]. From this perspective, further work will be necessary in this study to characterize the biopolymer synthetized by the two Gram-negative isolated strains to better envision its possible application.

#### 3.4.2. Biochemical Characterization

Using the Bergey’s Manual of Determinative Bacteriology and the API 20E system (Biomerieux), the two strains 1B and 2D1 were classified up to the genus level, based on morphological and biochemical properties (Figure 7). While 2D1 showed fermentation properties towards glucose, sucrose, lactose, and maltose, 1B was capable of fermenting the majority of the carbohydrates tested: Glucose, mannose, inositol, sorbitol, sucrose, amygdaline, arabinose, lactose, maltose, and fructose (Figure 7 and Figure 8). Both strains 1B and 2D1 were identified as oxidase negative (Figure 9), while only the 2D1 was urease positive. All the biochemical characteristics of 1B and 2D1 and provisional identification are reported in detail in Table 2. In a similar work, the same approach was used to identify up to 16 isolates. The work reported that morphological and biochemical tests identified the isolates as belonging to the genus *Bacillus*, *Micrococcus*, *Veillonella*, and *Pseudomonas* [19]. Similarly, in the work of Thapa et al. [23], the morphology and biochemical study identified the seven isolates from field soil, industrial sites, and animal manure belonging to the *Bacillus*, *Micrococcus*, and *Arthrobacter* genus [23]. In accordance with the morphological and biochemical characteristics reported in this work, the strain 1B belongs to the genus *Serratia*, while the strain 2D1 to the genus *Proteus*. In the literature, other studies report the use of *Serratia* sp. to synthetize and extract PHB [30]. Lugg et al. [30] showed that a strain of *Serratia* sp., showing intracellular bodies that are dark colored following the Sudan Black staining, was able to produce 55% of stored material per gram of biomass dry weight by consuming glycerol as the sole source of carbon and under nitrogen limitations. Again, Gupta et al. [31] confirmed the possibility to produce the biopolymer along with the wastewater treatment by the strain of *Serratia* sp. The work highlights how the biodegrading properties of the bacterial species for decontamination of the environment, as well as the valorization of wastewater produce the biopolymer [31]. Similarly, the genus *Proteus* in mixed cultures with *Enterobacter* and *Bacillus* lead to the final yield of 64.7% (*w*/*w*) of PHB in a batch fermentation process using coconut coir [32]. This promising study can prove the possibility of producing biopolymers from biowastes, as well as improving the microbial productivity within a combination of different bacterial cultures that enhance the final PHB yields.

#### 3.4.3. Antibiotic Susceptibility 

The antibiotic susceptibility test was implemented as an additional analysis for the identification of our PHB isolates, in order to confirm the genera our isolates belong to as determined by the morphological and biochemical tests. Of the fifteen antibiotics, for amoxicillin, cefepime, ampicillin/sulbactam, penicillin, imipenem, roxithromycin, and floxapen both strains showed resistance, while for chloramphenicol, kanamycin, nitroxoline, and flumequine they both showed susceptibility. The results of the antimicrobial tests for the other antibiotics are reported in Table 3. Unlike the results obtained in our study, the antimicrobial susceptibility tests conducted by Stock et al. [33] on *Serratia* sp., reported that the genus *Serratia* showed susceptibility to amoxicillin, ampicillin/sulbactam, penicillin, and amoxicillin/clavulanic acid [33]. Similarly, while in our analysis the strain identified as belonging to the genus *Proteus* showed resistance to amoxicillin, ampicillin/sulbactam, and imipenem, the studies conducted by Stock et al. [34] report that *Proteus* sp. are susceptible to these antibiotics [34]. This experimental evidence suggests that different species belonging to the same genus, both *Serratia* and *Proteus*, might show a widely different antimicrobial behavior. The additional antibiotic susceptibility tests performed in this study cannot confirm or support what is already identified through the morphological and biochemical analysis. In this perspective, it raises the need to conduct a molecular analysis of the isolated bacterial strains to confirm their identity.

## 4. Conclusions

The purpose of this work was to isolate and preliminarily identify bacteria able to synthetize the biopolymer polyhydroxybutyrate from argan crop soil in the southwestern region of Morocco, where the argan tree species *Argania spinosa* is limited to this geographical area of the country. Several soil samples were collected from different locations and pretreated by dilutions and culturing after thermal shock to select only heat resistant bacteria. Among all the samples analyzed, only four thermoresistant strains were isolated and studied for the presence of intracellular spores. Of the four isolated strains, only two, named 1B and 2D1, were identified as endospore positive strains by the Schaeffer-Fulton endospore staining. This first identification led to investigating more of the quality of the intracellular spores. The Sudan black stain is a lipophilic solution that has affinity for membranes and the biopolymer itself, hence endospores and PHB granules appear dark blue or black following the staining with Sudan black B. Indeed, black smears and granules were observed on petri dishes with the naked eye and on glass slides in microscopic observation, respectively, confirming that both strains, 1B and 2D1, are positive for the synthesis of PHB. Regular white colonies, translucent, with a smooth surface were the morphological characteristics that 1B and 2D1 had in common by observing each of the pure culture. Both strains were identified as Gram-negative bacteria, although 1B formed larger and clustered colonies compared to 2D1. Moreover, the use of biochemical tests identified the 1B strain as able to ferment more carbohydrates compared to 2D1, while the latter was tested as urease positive unlike 1B. Based on the Gram staining, morphological characteristics, and biochemical tests, the two strains, 1B and 2D1, were identified as belonging to the genus *Serratia* and *Proteus*, respectively. Nonetheless, the results of the antimicrobial susceptibility tests differ greatly from what was reported in the literature and, consequently, the results of this analysis could not confirm what was previously identified through the morphological and biochemical studies.

Hence, this work aimed to isolate and preliminarily identify, for the first time, bacterial species with PHB producing ability from the argan crop soil in Morocco. The two isolates identified as belonging to the genera *Serratia* and *Proteus* may have promising genetic-molecular characteristics for the production, extraction, and processing of PHB using argan fruit and argan oil pressing wastes. 

## Figures and Tables

**Figure 1 polymers-13-01870-f001:**
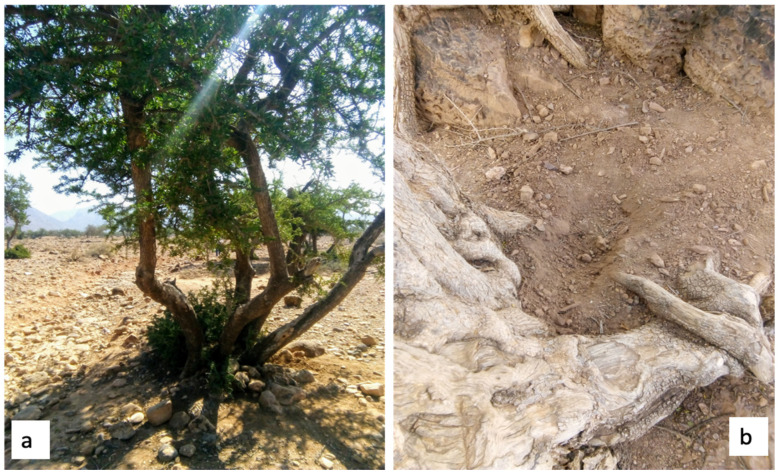
(**a**) *Argania spinosa* tree in the southwestern region of Morocco, and (**b**) one of the rizospheric soil.

**Figure 2 polymers-13-01870-f002:**
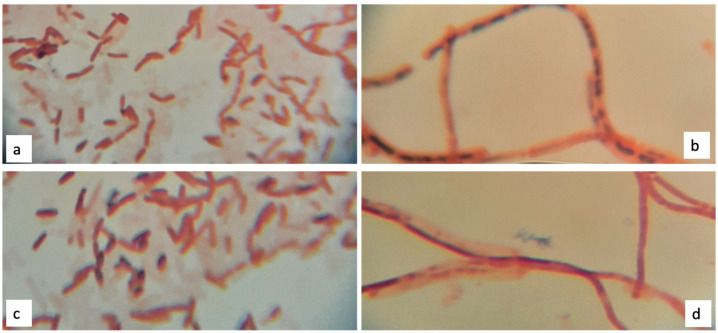
Schaeffer-Fulton endospore staining with Malachite green (**a**,**b**) and Methylene blue (**c**,**d**); presence of green and blue granules, respectively, for the strain 1B (**a**,**c**) and 2D1 (**b**,**d**).

**Figure 3 polymers-13-01870-f003:**
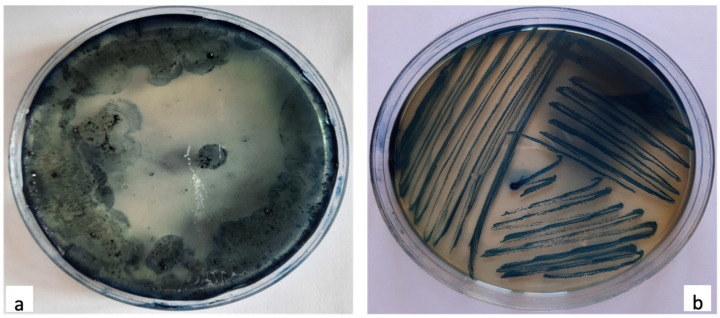
Sudan black B staining for PHB smears identification on petri dishes; presence of black smears for the two positive PHB producers: 1B (**a**) and 2D1 (**b**).

**Figure 4 polymers-13-01870-f004:**
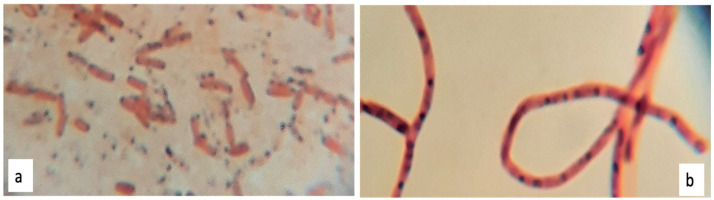
Sudan black staining for PHB granules identification on glass slides; presence of black granules for the two positive PHB producers: 1B (**a**) and 2D1 (**b**).

**Figure 5 polymers-13-01870-f005:**
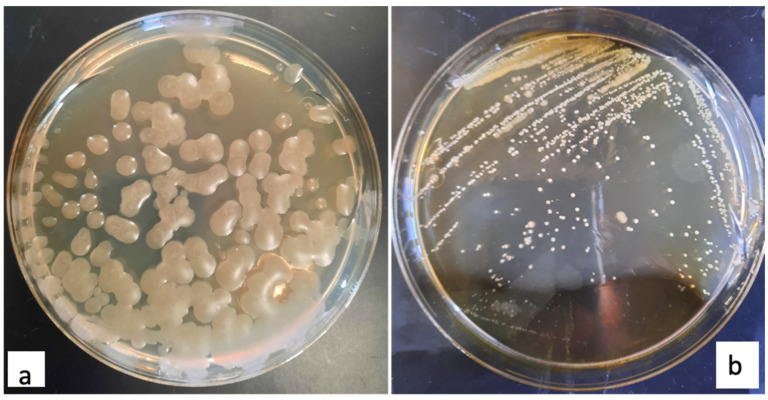
Pure cultures of isolated bacteria (**a**) 1B and (**b**) 2D1 resulted as PHB positive.

**Figure 6 polymers-13-01870-f006:**
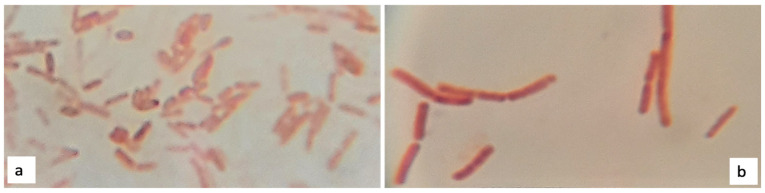
Gram staining: 1B (**a**) and 2D1 (**b**) appear colored in red, hence classified as Gram-negative bacteria.

**Figure 7 polymers-13-01870-f007:**
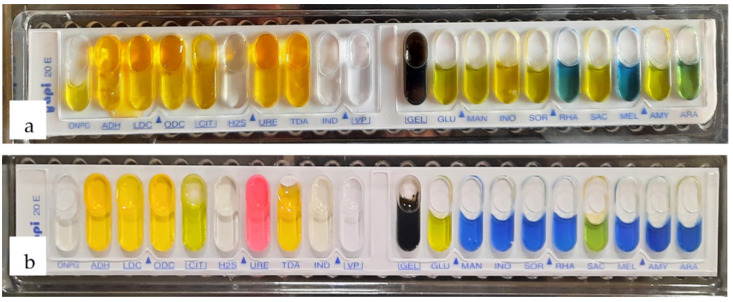
Biochemical tests API 20E Enterobacteriaceae for (**a**) 1B and (**b**) 2D1.

**Figure 8 polymers-13-01870-f008:**
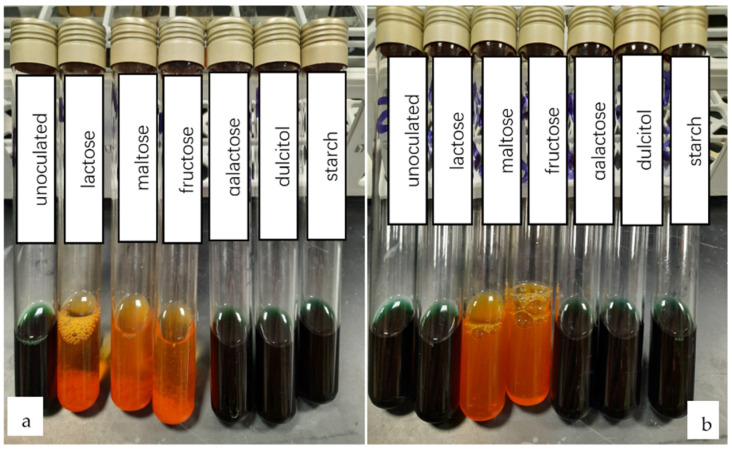
Sugar fermentation tests using lactose, maltose, fructose, galactose, dulcitol, and starch for (**a**) 1B and (**b**) 2D1 strains.

**Figure 9 polymers-13-01870-f009:**
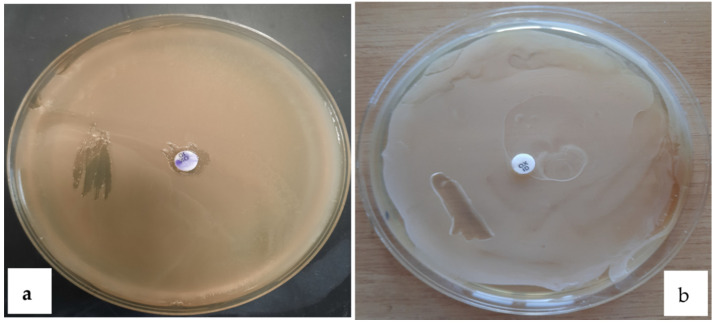
Sugar fermentation tests using lactose, maltose, fructose, galactose, dulcitol, and starch for (**a**) 1B and (**b**) 2D1 strains.

**Table 1 polymers-13-01870-t001:** Morphological characteristics of the strains 1B and 2D1.

Morphology	1B Strain	2D1 Strain
shape	regular, entire, spherical, large	regular, entire, sherical, small
margin	regular	regular
color	white	white
opacity	translucent	translucent
elevation	convex	droplike
size (diameter)	3–5 mm	0.5–1 mm
surface	smooth, glisterine	smooth, glisterine
texture	moist, viscid	dry
arrangement	clustered	isolated

**Table 2 polymers-13-01870-t002:** Provisional identification of the PHB producers 1B and 2D1.

Biochemical Tests	1B Strain	2D1 Strain
beta galactosidase	+	-
arginine decarboxylase	-	-
lysine decarboxylase	-	-
ornithine decarboxylase	-	-
citrate	-	-
hydrogen sulfite production	-	-
urease	-	+
tryptophane deaminase	-	-
indole test	-	-
Voges Proskauer	-	-
gelatinase presence	+	+
glucose fermentation	+	+
mannose fermentation	+	-
inositol fermentation	+	-
sorbitol fermentation	+	-
rhamnose fermentation	-	-
sucrose fermentation	+	+
melibiose fermentation	-	-
amygdalin fermentation	+	-
arabinose fermentation	+	-
oxidase test	-	-
lactose	+	-
maltose	+	+
fructose	+	+
galactose	-	-
dulcitol	-	-
starch	-	-
Probable genus	*Serratia*	*Proteus*

**Table 3 polymers-13-01870-t003:** Antimicrobial susceptibility of the PHB producers 1B and 2D1.

Antibiotic	Symbol	Concentration (μg)	1B Genus *Serratia*	2D1 Genus *Proteus*
amoxicillin	AML	2	resistant	resistant
cefepime	FEP	30	resistant	resistant
ampic/sulbac	SAM	20	resistant	resistant
penicilline G	P	10	resistant	resistant
imipenem	IPM	10	resistant	resistant
roxithromocyn	RXT	15	resistant	resistant
floxapen	FU	5	resistant	resistant
gentamicin	CN	10	intermediate	intermediate
chloramphenicol	C	30	susceptible	susceptible
bacitracin	B	10	resistant	intermediate
ampic/clav acid	AMC	30	resistant	intermediate
kanamycin	K	30	susceptible	susceptible
nitroxoline	NI	20	susceptible	susceptible
flumequine	AR	30	susceptible	susceptible
teicoplanin	TEC	30	resistant	intermediate

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
