# Peer review of "Isolation of Two Bacterial Species from Argan Soil in Morocco Associated with Polyhydroxybutyrate (PHB) Accumulation: Current Potential and Future Prospects for the Bio-Based Polymer Production"

_polymers, 2021, doi:10.3390/polym13111870_

Round 1
Reviewer 1 Report
I read carefully the manuscript entitled Isolation and Characterization of PHB Producing Isolates From Argan Soil in Morocco.
This manuscript contains very preliminary results. In this study authors have isolated the PHA producing strains and studied biochemical test and staining. There is no data of polymers production without this it seems difficult to publish in a reputed Polymers Journal.
Authors are encouraged to resubmit by incorporating PHB production data and for this refer some recent studies where isolated strains showed ability to produce PHB is an effective manner.
Author Response
Reviewer I
I read carefully the manuscript entitled Isolation and Characterization of PHB Producing Isolates From Argan Soil in Morocco.
This manuscript contains very preliminary results. In this study authors have isolated the PHA producing strains and studied biochemical test and staining. There is no data of polymers production without this it seems difficult to publish in a reputed Polymers Journal.
Authors are encouraged to resubmit by incorporating PHB production data and for this refer some recent studies where isolated strains showed ability to produce PHB is an effective manner.
First of all, we would thank the Editor for giving us the possibility to improve our manuscript. We would also express our gratitude to the Reviewers for their valuable work, constructive comments and thoughtful suggestions.
Indeed, as you previously mentioned in your comments, our manuscript contains preliminary results. Our work introduces the beginning of a research aimed to isolate and identify PHB-producing bacteria for the first time from argan crops; produce and characterize PHB using argan wastes; and determine the biodegradability of the biopolymer synthesized. In this manuscript it is reported the first part of our work since we potentially identified bacterial species from argan soil, able to synthesize PHB. As you suggested, our following work will be the production of the biopolymer using argan fruits and press cake wastes, and its characterization. As you suggested, at that point we will have the PHB production data to effectively support our hypothesis of “PHB-producing bacteria from argan soil using argan wastes”.

Reviewer 2 Report
The manuscript has several shortcomings. After editing and correcting the shortcomings, it will be possible to consider publishing it in the journal Polymers.
Personally, I miss a substantive discussion with other researchers. It would constitute a better foundation for the conducted research.
The chapter "conclusions" provides a summary. The section "conclusions" is not of the nature of conclusions.
Author Response
Reviewer II
The manuscript has several shortcomings. After editing and correcting the shortcomings, it will be possible to consider publishing it in the journal Polymers.
Personally, I miss a substantive discussion with other researchers. It would constitute a better foundation for the conducted research.
The chapter "conclusions" provides a summary. The section "conclusions" is not of the nature of conclusions.
First of all, we would thank the Editor for giving us the possibility to improve our manuscript. We would also express our gratitude to the Reviewers for their valuable work, constructive comments and thoughtful suggestions. Based on these comments and suggestions, we have made the modifications on the original manuscript.
Additional paragraphs have been added in the introduction section to better contextualize the role of PHAs/PHB as alternative to conventional plastics, and to report more about isolated PHB producers form agriculture crops. The objectives of the work have been better organized in a bullet form at the end of the introduction to widely specify each aim and its significance. Some minor changes have been made for “Materials and Methods”. Some minor changes have been made in the section “Results and Discussion”. A better presentation of the antibiotic susceptibility tests has been made to give relevance and significance of the results obtained. The final paragraph of the “Conclusion” section has been modified to remove redundant elements already presented in the previous sections or unnecessary additional elements of future works. The “References” have been standardized. The title has been aligned with the aims of the work.

Reviewer 3 Report
Brief summary
The manuscript "Isolation and Characterization of PHB Producing Isolates from Argan Soil in Morocco” aimed to isolate and identify PHB-producing bacteria from the agricultural soil samples of Argania spinosa in the south region of Morocco. I believe this study is a contribution to assist in understanding of the identification of bacterial species producing PHB, and influence of specific species and substrates on PHB quality. Overall, the provided information is a good resource; however, unfortunately the authors have not fully developed some topics of the manuscript and I have decided that the manuscript, in this version, needs a substantial revision.
Broad comments to authors:
Overall, I recommend clarifying some aspects:
- The results are not aligned with the title and aims, as well as the development. It seems to be much broader.
- Contextualization of the introduction and objective (for example, some aspects presented in the objectives and results were not widely contextualized in the introduction); of the methods in order to allow results interpretation; and discussion, which should be further expanded and justified.
- It’s unclear what questions you are asking, why it’s interesting, why isolation and characterization of PHB can be interesting to be researched in an argan soil context. In the introduction, for example, leave your goals in topic style (i, ii, iii ...) this can facilitate understanding.
- I believe that the current structure of the manuscript makes it difficult to understand, I think. I think the manuscript needs a much clearer structure that is closely linked to questions (which need to be defined much more clearly, please see earlier comments), and the results of those questions. There are lot of information and ideas (mainly in the results, which I don't see in the objectives), but I’m struggling to see how it all fits together, and how it fits with the data collected, and what the story is.
For example, it is not possible to see in the text an appropriate contextualization on polyhydroxyalkanoates (PHAs)/ polyhydroxybutyrate (PHB), as being a class of polymers that can serve as potential substitutes for conventional plastics (including introduction and discussion - and international literature), regarding gaps and the importance of methodological aspects. Lack of alignment of the title with objective and sequence of methodological procedures and results, this also reflects in the structure of the discussion.
Many problems are detected in the formatting of the article, including the standardization of citations throughout the text.
These are the main problems I found in the manuscript, but there are many other minor mistakes or suggestions I indicated below, and I hope they may help the authors when reviewing their work. The detailed suggestions follow below.
Specific comments to authors:
Title:
I have a reservation as to the title of the work.
Point 1: Make it clear what the acronym is about, for example: polyhydroxybutyrate (PHB)
Point 2: from NOT From
Point 3: What is new about this study in relation to the article? Aragosa, A. et al. (2020). PHB Produced by Bacteria Present in the Argan Field Soil: A New Perspective for the Synthesis of the Bio-Based Polymer. Proceedings 2021, 69, 5. https://doi.org/10.3390/CGPM2020-07226
Introduction:
Point 1, General: The introduction of the work should better contextualize the "polyhydroxyalkanoates (PHAs) polyhydroxybutyrate (PHB), as being a class of polymers that can serve as potential substitutes for conventional plastics".
For example, see:
Gülsah Keskin, Gülnur Kızıl, Mikhael Bechelany, Céline Pochat-Bohatier and Mualla Öner. (2017). Potential of polyhydroxyalkanoate (PHA) polymers family as substitutes of petroleum based polymers for packaging applications and solutions brought by their composites to form barrier materials. Pure and Applied Chemistry, 89(12). DOI: https://doi.org/10.1515/pac-2017-0401
Schlebrowski, T.; Ouali, R.; Hahn, B.; Wehner, S.; Fischer, C.B. Comparing the Influence of Residual Stress on Composite Materials Made of Polyhydroxybutyrate (PHB) and Amorphous Hydrogenated Carbon (a-C:H) Layers: Differences Caused by Single Side and Full Substrate Film Attachment during Plasma Coating. Polymers 2021, 13, 184. https://doi.org/10.3390/polym13020184
Al Battashi, H., Al-Kindi, S., Gupta, V.K. et al. Polyhydroxyalkanoate (PHA) Production Using Volatile Fatty Acids Derived from the Anaerobic Digestion of Waste Paper. J Polym Environ 29, 250–259 (2021). https://doi.org/10.1007/s10924-020-01870-0
Point 2; Line 48: polyhydroxybutyrate (PHB)
Point 3; Lines 48-50: Before PHB I suggest doing a brief contextualization of polyhydroxyalkanoates (PHAs), as being a class of polymers that can serve as potential substitutes for conventional plastics.
Point 3; Line 64: Thammasittirong et al. [6], reported that…
Point 4; Line 73: Chandani Devi et al. [7], used the…
Point 5; Line 73: et al. NOT et. al.
Point 6; Line 100: "and" without italics
Point 6; Line 118: Insert space between “Rhodococcus equi” and “produces”
Point 7; Line 125: Bacillus sp. NOT Bacillus sp.
Point 8; Lines 129-136: The objectives need to be clear/aligned with what has been done. Perhaps in the form of dots: (i) presence; (ii) abundance; etc.
Materials and Methods:
Point 1; Line 141: Make it clear what sterile conditions are.
Point 2; Lines 225-239: At no time is there mention of susceptibility to antibiotics, for example, in the context of the introduction and objectives. If one of the objectives of the work is to assess susceptibility to antibiotics, this needs to be stated in the other parts of the manuscript.
Point 3; Line 239: Did the authors proceed with any statistics for data analysis?
Results:
General point [discussion]: The results of research on the PHB-producing bacteria from the agricultural soil samples of Argania spinosa are interesting, but based on too few samplings. However, they were not related to similar studies described in the world literature. The literature review should be enriched with studies on the PHB-producing bacteria. There is no discussion section that relates your findings with relevant scientific background. Once you prepare a real introduction section, it will also help you to make up a much better discussion section.
Point 1; Line 241: Wouldn't it be Results and Discussion?
Point 2; Lines 244-247: These are methods. Repetitive.
Point 3; Lines 249-251: This also seems to me to be characteristic of methodology. Am I right?
Point 4; Lines 250: Wasn't it in the crops?
Point 5; Figure 1: This figure is much more appropriate for the methodology.
Point 6; Lines 262-264: These are methods. Repetitive.
Point 7; Line 264: Repeated endpoint.
Point 8; Line 290: Result of this study or the literature?
Point 9; Line 290-292: Start with the results of this study, then discuss.
Point 10; Line 321: Period in missing sentence.
Point 11; Table 1: This is a frame and not a table. Use editor to make it as a table and not as a figure.
Point 12; Line 377: Year of publication, for example, Lugg et al. [25]…
Point 13; Line 378: Serratia sp. NOT Serratia sp.
Point 14; Line 381: Year of publication, for example, Gupta et al. [26],…
Point 15; Line 382: Serratia sp. NOT Serratia sp..
Point 16; Line 385: Proteus NOT Proteus; Enterobacter and Bacillus NOT Enterobacter and Bacillus
Point 17; Line 433: Year of publication, for example, Stock et al. [28]…
Point 19; Line 439: Serratia than Proteus NOT Serratia than Proteus
Point 19; Table 2: This is a frame and not a table. Use editor to make it as a table and not as a figure.
Point 20; Lines 427-441: What did this contribute to your study?
Point 21; Table 3: This is a frame and not a table. Use editor to make it as a table and not as a figure.
Conclusion:
General point: The final conclusions are limited to the described case only, there is no discussion and comparison with similar studies described in the literature. In other words, there are no conclusions about methodological aspects that could improve the interest of the paper. All the conclusions are too local and focused in the problems of a specific Morocco area.
Point 1; Lines 452-454: These are methods. Repetitive.
Point 2; Lines 458-460: It seems the conclusion of a work that aimed to evaluate better methods. Review.
Point 3; Lines 460-463: Is that a conclusion or introduction?
Point 4; Line 474: Serratia and Proteus NOT Serratia and Proteus
Point 5; Lines 475-476: This is objective and not a conclusion. Repetitive.
Point 6; Lines 476-478: Take to the beginning of the conclusions.
Point 7: Is there no conclusion for Antibiotic susceptibility.
References:
General: Your references (all) are not standardized correctly. Please check the "Instructions for Authors": https://www.mdpi.com/journal/polymers/instructions
Author Response
Reviewer III
First of all, we would thank the Editor for giving us the possibility to improve our manuscript. We would also express our gratitude to the Reviewers for their valuable work, constructive comments and thoughtful suggestions. Based on these comments and suggestions, we have made the modifications on the original manuscript. Responses to the comments with description of the revisions implemented in the manuscript are following detailed:
Brief summary
The manuscript "Isolation and Characterization of PHB Producing Isolates from Argan Soil in Morocco” aimed to isolate and identify PHB-producing bacteria from the agricultural soil samples of Argania spinosa in the south region of Morocco. I believe this study is a contribution to assist in understanding of the identification of bacterial species producing PHB, and influence of specific species and substrates on PHB quality. Overall, the provided information is a good resource; however, unfortunately the authors have not fully developed some topics of the manuscript and I have decided that the manuscript, in this version, needs a substantial revision.
Broad comments to authors:
Overall, I recommend clarifying some aspects:
- The results are not aligned with the title and aims, as well as the development. It seems to be much broader.
- Contextualization of the introduction and objective (for example, some aspects presented in the objectives and results were not widely contextualized in the introduction); of the methods in order to allow results interpretation; and discussion, which should be further expanded and justified.
- It’s unclear what questions you are asking, why it’s interesting, why isolation and characterization of PHB can be interesting to be researched in an argan soil context. In the introduction, for example, leave your goals in topic style (i, ii, iii ...) this can facilitate understanding.
- I believe that the current structure of the manuscript makes it difficult to understand, I think. I think the manuscript needs a much clearer structure that is closely linked to questions (which need to be defined much more clearly, please see earlier comments), and the results of those questions. There are lot of information and ideas (mainly in the results, which I don't see in the objectives), but I’m struggling to see how it all fits together, and how it fits with the data collected, and what the story is.
For example, it is not possible to see in the text an appropriate contextualization on polyhydroxyalkanoates (PHAs)/ polyhydroxybutyrate (PHB), as being a class of polymers that can serve as potential substitutes for conventional plastics (including introduction and discussion - and international literature), regarding gaps and the importance of methodological aspects. Lack of alignment of the title with objective and sequence of methodological procedures and results, this also reflects in the structure of the discussion.
Many problems are detected in the formatting of the article, including the standardization of citations throughout the text.
These are the main problems I found in the manuscript, but there are many other minor mistakes or suggestions I indicated below, and I hope they may help the authors when reviewing their work. The detailed suggestions follow below.
After reading your initial comments, we changed the manuscript according to the specific comments you listed below, so each point is commented/answered as follow and the changes are made on the manuscript using the “track changes”
Specific comments to authors:
Title:
I have a reservation as to the title of the work.
we changed the title with a more consistent one; however, alternative titles are also listed in the manuscript “track changes”
Point 1: Make it clear what the acronym is about, for example: polyhydroxybutyrate (PHB)
DONE
Point 2: from NOT From
DONE
Point 3: What is new about this study in relation to the article? Aragosa, A. et al. (2020). PHB Produced by Bacteria Present in the Argan Field Soil: A New Perspective for the Synthesis of the Bio-Based Polymer. Proceedings 2021, 69, 5. https://doi.org/10.3390/CGPM2020-07226
The previous work was an initial isolation of thermoresistant species from argan crops, and only one species was identified, through Shaeffer-Foulton staining, as an eventual PHB producer. The current study includes: i) one more species has been isolated as potential PHB producer; ii) preliminary identification of both species using: staining procedures (including the Sudan black staining specific for PHB not used in the previous work), morphological and gram staining analysis, biochemical and sugar fermentation tests, and antibiotic susceptibility tests.
Introduction:
Point 1, General: The introduction of the work should better contextualize the "polyhydroxyalkanoates (PHAs) polyhydroxybutyrate (PHB), as being a class of polymers that can serve as potential substitutes for conventional plastics".
A second paragraph has been added to the “introduction” section to introduce PHAs
For example, see:
Gülsah Keskin, Gülnur Kızıl, Mikhael Bechelany, Céline Pochat-Bohatier and Mualla Öner. (2017). Potential of polyhydroxyalkanoate (PHA) polymers family as substitutes of petroleum based polymers for packaging applications and solutions brought by their composites to form barrier materials. Pure and Applied Chemistry, 89(12). DOI: https://doi.org/10.1515/pac-2017-0401
Schlebrowski, T.; Ouali, R.; Hahn, B.; Wehner, S.; Fischer, C.B. Comparing the Influence of Residual Stress on Composite Materials Made of Polyhydroxybutyrate (PHB) and Amorphous Hydrogenated Carbon (a-C:H) Layers: Differences Caused by Single Side and Full Substrate Film Attachment during Plasma Coating. Polymers 2021, 13, 184. https://doi.org/10.3390/polym13020184
Al Battashi, H., Al-Kindi, S., Gupta, V.K. et al. Polyhydroxyalkanoate (PHA) Production Using Volatile Fatty Acids Derived from the Anaerobic Digestion of Waste Paper. J Polym Environ 29, 250–259 (2021). https://doi.org/10.1007/s10924-020-01870-0
The suggested articles listed above have been included in the “introduction” section, and the references listed at the end of the manuscript in the “references” section
Point 2; Line 48: polyhydroxybutyrate (PHB)
DONE
Point 3; Lines 48-50: Before PHB I suggest doing a brief contextualization of polyhydroxyalkanoates (PHAs), as being a class of polymers that can serve as potential substitutes for conventional plastics.
A second paragraph has been added to the “introduction” section to introduce PHAs
Point 3; Line 64: Thammasittirong et al. [6], reported that…
DONE
Point 4; Line 73: Chandani Devi et al. [7], used the…
DONE
Point 5; Line 73: et al. NOT et. al.
DONE
Point 6; Line 100: "and" without italics
DONE
Point 6; Line 118: Insert space between “Rhodococcus equi” and “produces”
DONE
Point 7; Line 125: Bacillus sp. NOT Bacillus sp.
DONE
Point 8; Lines 129-136: The objectives need to be clear/aligned with what has been done. Perhaps in the form of dots: (i) presence; (ii) abundance; etc.
As requested, the objectives of the work have been added and organized in a dot form as you suggested; the changes are in the last paragraph of the “introduction” section
Moreover, we would like to communicate to you that while more information/additional paragraphs have been added in the “introduction” section, none of the previous paragraphs have been delated, eventually making this section too long for an introduction.
Materials and Methods:
Point 1; Line 141: Make it clear what sterile conditions are.
DONE
Point 2; Lines 225-239: At no time is there mention of susceptibility to antibiotics, for example, in the context of the introduction and objectives. If one of the objectives of the work is to assess susceptibility to antibiotics, this needs to be stated in the other parts of the manuscript.
the antibiotic susceptibility test is amply discussed throughout the manuscript: in the results/discussion, conclusion, abstract, to better contextualize the reason why such analysis has been implemented
Point 3; Line 239: Did the authors proceed with any statistics for data analysis?
Our preliminary work is a qualitative study aimed to isolate potential PHB producers, and to preliminarly identify the genus the two isolated species belong to. PHB production and data analysis will be reported in a further work when PHB production and characterization will be determined using argan fruit and press cake wastes
Results:
General point [discussion]: The results of research on the PHB-producing bacteria from the agricultural soil samples of Argania spinosa are interesting, but based on too few samplings. However, they were not related to similar studies described in the world literature. The literature review should be enriched with studies on the PHB-producing bacteria. There is no discussion section that relates your findings with relevant scientific background. Once you prepare a real introduction section, it will also help you to make up a much better discussion section.
A paragraph has been added to the “introduction” section (lines 108-120) to better contextualize PHB produced by bacteria isolated from soil samples. As you observed, more samples and species have been isolated from other works in the literature. We perfectly admit this observation. However, although our initial study is based on few samples collected (a total of 40 soil samples from different locations) and few strains isolated as potential PHB producers (2 species), we firmly confirm our promising and innovating research, which for the first time is looking for PHB-producing bacteria in argan soil. With the same conviction we admit our audacity in attempting to present such a preliminary work.
Point 1; Line 241: Wouldn't it be Results and Discussion?
DONE
Point 2; Lines 244-247: These are methods. Repetitive.
DONE
Point 3; Lines 249-251: This also seems to me to be characteristic of methodology. Am I right?
DONE
Point 4; Lines 250: Wasn't it in the crops?
DONE
Point 5; Figure 1: This figure is much more appropriate for the methodology.
DONE
Point 6; Lines 262-264: These are methods. Repetitive.
DONE
Point 7; Line 264: Repeated endpoint.
DONE
Point 8; Line 290: Result of this study or the literature?
DONE
Point 9; Line 290-292: Start with the results of this study, then discuss.
DONE
Point 10; Line 321: Period in missing sentence.
DONE
Point 11; Table 1: This is a frame and not a table. Use editor to make it as a table and not as a figure.
We tried to fix the frame and replace it as a table and/or to make a new table again; however, once we edit as a table or create a new table it doesn’t allow us to change font/color/size; would you please suggest us how this can be done? At the moment we left a table which is not designed in the way we want it.
Point 12; Line 377: Year of publication, for example, Lugg et al. [25]…
DONE
Point 13; Line 378: Serratia sp. NOT Serratia sp.
DONE
Point 14; Line 381: Year of publication, for example, Gupta et al. [26],…
DONE
Point 15; Line 382: Serratia sp. NOT Serratia sp..
DONE
Point 16; Line 385: Proteus NOT Proteus; Enterobacter and Bacillus NOT Enterobacter and Bacillus
DONE
Point 17; Line 433: Year of publication, for example, Stock et al. [28]…
DONE
Point 19; Line 439: Serratia than Proteus NOT Serratia than Proteus
DONE
Point 19; Table 2: This is a frame and not a table. Use editor to make it as a table and not as a figure.
We tried to fix the frame and replace it as a table and/or to make a new table again; however, once we edit as a table or create a new table it doesn’t allow us to change font/color/size; would you please suggest us how this can be done? At the moment we left a table which is not designed in the way we want it.
Point 20; Lines 427-441: What did this contribute to your study?
Our work was implemented to preliminarily identify the two isolated species, and to determine the genera they might belong to. The use of morphological analysis, Gram staining, biochemical tests and antibiotic susceptibility tests represent a wide range of tests to determine the genera of the two strains. Including the antibiotic susceptibility tests was used to confirm the reliability of all previous results, although, as reported in the discussion, our antibiotic susceptibility results couldn’t confirm that. Of course, we are aware of the need to add molecular analysis tests, not only to confirm the genus but also to define the species. Unfortunately, at the moment, such analysis was not possible to be included in this manuscript.
Results and discussion for the antibiotic susceptibility analysis has been modified to better specify the necessity to include such additional test.
Point 21; Table 3: This is a frame and not a table. Use editor to make it as a table and not as a figure.
We tried to fix the frame and replace it as a table and/or to make a new table again; however, once we edit as a table or create a new table it doesn’t allow us to change font/color/size; would you please suggest us how this can be done? At the moment we left a table which is not designed in the way we want it.
Conclusion:
General point: The final conclusions are limited to the described case only, there is no discussion and comparison with similar studies described in the literature. In other words, there are no conclusions about methodological aspects that could improve the interest of the paper. All the conclusions are too local and focused in the problems of a specific Morocco area.
We modified the conclusion by adding the antibiotic susceptibility tests; we modified the last paragraph by deleting further methodology and/or work; however, we wanted to state at the end the importance of isolating such types of strains from argan soil for the first time. We wanted to point out the importance of this initial work as potentially innovating in the production and transformation of PHB by using the argan wastes. That is fundamentally the reason why we started isolating bacterial species from argan crops. Otherwise, whether this paragraph might be considered inconsistent to the “conclusion” section, it will be deleted at your convenience.
Point 1; Lines 452-454: These are methods. Repetitive.
DONE
Point 2; Lines 458-460: It seems the conclusion of a work that aimed to evaluate better methods. Review.
I am not sure I fully understand your comment here; however, I tried to delete future prospects
Point 3; Lines 460-463: Is that a conclusion or introduction?
A sentence has been deleted
Point 4; Line 474: Serratia and Proteus NOT Serratia and Proteus
DONE
Point 5; Lines 475-476: This is objective and not a conclusion. Repetitive.
DONE
Point 6; Lines 476-478: Take to the beginning of the conclusions.
DONE
Point 7: Is there no conclusion for Antibiotic susceptibility.
A paragraph for the antibiotic susceptibility has been added
References:
General: Your references (all) are not standardized correctly. Please check the "Instructions for Authors": https://www.mdpi.com/journal/polymers/instructions
DONE

Round 2
Reviewer 1 Report
Authors have not performed and included the results of PHB production and characterization studies. Only isolation and identification of PHB producing strain is not sufficient to publish the research work in Polymers.
Considering this I am rejecting this manuscript to publish in Polymers..
Reviewer 3 Report
Brief summary and broad comments to authors:
The manuscript now titled "Isolation of Two Bacterial Species from Argan Soil in Morocco Associated with Polyhydroxybutyrate (PHB) Accumulation: Current Potential and Future Prospects for the Bio-Based Polymer Production” aimed to isolate and identify PHB-producing bacteria from the agricultural soil samples of Argania spinosa in the south region of Morocco. I believe this study is a contribution to assist in understanding of the identification of bacterial species producing PHB, and influence of specific species and substrates on PHB quality. Overall, the provided information is a good resource. The authors have made efforts to amend the manuscript based upon the original comments. The changes made by the authors improve the manuscript. On balance, I believe this manuscript is now suitable for publication.
Some details:
Line 43: I believe there is a surplus space
Line 108: Line: I believe that there is a surplus space after Lathwal et al. [14]
Line 110: "and" without italics
Line 161: I believe there is a surplus space
Line 166: delete the “t”
Lines 171, 378, 432: I believe there is a surplus space